# A cryptic pocket in Ebola VP35 allosterically controls RNA binding

Matthew A. Cruz [1,4], Thomas E. Frederick[1,4], Upasana L. Mallimadugula [1], Sukrit Singh [1], Neha Vithani[1], Maxwell I. Zimmerman[1], Justin R. Porter [1], Katelyn E. Moeder[1], Gaya K. Amarasinghe [2] & Gregory R. Bowman [1,3✉]

Protein-protein and protein-nucleic acid interactions are often considered difficult drug targets because the surfaces involved lack obvious druggable pockets. Cryptic pockets could present opportunities for targeting these interactions, but identifying and exploiting these pockets remains challenging. Here, we apply a general pipeline for identifying cryptic pockets to the interferon inhibitory domain (IID) of Ebola virus viral protein 35 (VP35). VP35 plays multiple essential roles in Ebola's replication cycle but lacks pockets that present obvious utility for drug design. Using adaptive sampling simulations and machine learning algorithms, we predict VP35 harbors a cryptic pocket that is allosterically coupled to a key dsRNA-binding interface. Thiol labeling experiments corroborate the predicted pocket and mutating the predicted allosteric network supports our model of allostery. Finally, covalent modifications that mimic drug binding allosterically disrupt dsRNA binding that is essential for immune evasion. Based on these results, we expect this pipeline will be applicable to other proteins.

[1] Department of Biochemistry & Molecular Biophysics, Washington University School of Medicine, St. Louis, MO 63110, USA. [2] Department of Pathology & Immunology, Washington University School of Medicine, St. Louis, MO 63110, USA. [3] Center for the Science and Engineering of Living Systems, Washington University in St. Louis, St. Louis, MO 63110, USA. [4] These authors contributed equally: Matthew A. Cruz, Thomas E. Frederick. ✉email: g.bowman@wustl.edu

Examining structures available in the protein data bank (PDB) suggests that many protein surfaces that engage in protein–protein interactions (PPIs) and protein–nucleic acid interactions (PNIs) lack druggable pockets[1,2]. As a result, PPIs and PNIs are often considered intractable drug targets even when there is strong evidence that disrupting these interactions would be of great therapeutic value[3].

Cryptic pockets present opportunities for designing drugs for difficult targets like PPIs and PNIs but identifying and exploiting these pockets remains challenging[4–6]. Cryptic pockets are absent in available experimental structures but form in a subset of excited states that arise due to protein dynamics. These cryptic sites can serve as valuable drug targets if they coincide with key functional sites, or if they are allosterically coupled to distant functional sites[7,8]. Most known cryptic sites were only identified after the serendipitous discovery of a small molecule that binds and stabilizes the open form of the pocket[8,9]. Unfortunately, we currently lack methodology that can decouple pocket discovery from ligand discovery. To overcome this limitation and to increase the number of druggable targets, we have developed a suite of computational and experimental methods for detecting cryptic pockets and allostery, in addition to other available approaches[8,10–23]. We have successfully applied subsets of this toolset to a number of enzymes that are established drug targets[12,24], suggesting that the same tools may be ready for application to challenging targets like PPIs and PNIs.

Here, we present the first integration of our entire pipeline of tools to hunt for cryptic pockets in a difficult, non-enzymatic target that engages in PPIs and PNIs: the interferon inhibitory domain (IID) of Ebola viral protein 35 (VP35). Ebola virus causes a hemorrhagic fever that is often lethal, with case fatality rates approaching 90% in past outbreaks[25,26]. Initial promising results with the antiviral, remdesivir fell short in a randomized controlled trial so there remains no approved small-molecule drugs for treating Ebola[27]. Small-molecule antivirals are needed despite recent progress with antibodies[27] because they offer many advantages, including ease of delivery, lower cost, and longer shelf life that are particularly relevant in rural and impoverished regions. The ~120 residue IID of VP35 would be an appealing drug target for combating Ebola and other viruses in the *Filoviridae* family apart from lacking obvious druggable sites that could disrupt its PPI and PNIs. VP35 has a well-conserved sequence and plays multiple essential roles in the viral replication cycle[28]. One of its primary functions is to antagonize the host's innate immunity, particularly RIG-I-like receptor (RLR)-mediated detection of viral nucleic acids, to prevent an interferon (IFN) response and signaling of neighboring cells to heighten their antiviral defenses[29–31].

Crystal structures have revealed that VP35's IID binds both the blunt ends and backbone of double-stranded RNA (dsRNA), and that there is a PPI between these dsRNA-binding modes (Fig. 1)[32,33]. Binding to dsRNA blunt ends plays a dominant role in IFN suppression by Ebola[34]. Indeed, mutations that reduce the IID's affinity for dsRNA blunt ends are sufficient to mitigate IFN antagonism, ultimately attenuating Ebola's pathogenicity[34–37]. Therefore, disrupting this single binding mode could dramatically reduce the impact of an Ebola infection on the host and potentially reduce deleterious effects, including lethality. However, both dsRNA-binding interfaces are large flat surfaces that are difficult for small molecules to bind tightly (Fig. 1). As a result, only a few studies have sought to find small molecules targeting VP35, none of which has evolved into a full drug-discovery campaign[38–41]. The discovery of cryptic pockets in VP35 could provide new opportunities for drugging this essential viral component.

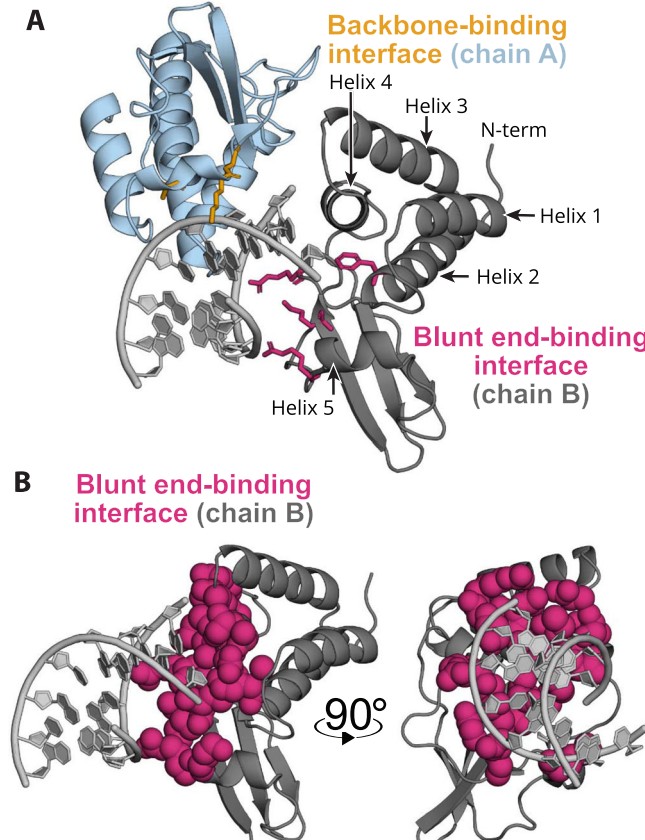

**Fig. 1 VP35 dsRNA interactions occur primarily through flat interfaces.**
**A** Crystal structure of two copies of VP35's IID (dark gray and light blue) bound to dsRNA (light gray) via two flat interfaces (PDB ID 3L25). Mutations to residues highlighted in pink and yellow sticks eliminate dsRNA binding. **B** Isolated chain B from the same view as panel A and after 90° rotation in the Y axis now highlighting the dsRNA interacting VP35 surface in the blunt-end-binding protomer. The blunt-end-binding interface (pink, 3L25 chain B) is shown as spheres to highlight that VP35 lacks deep pockets amenable to binding small molecules.

## Results

**Adaptive sampling simulations reveal a potentially druggable cryptic pocket**. To discover structures with large pocket volumes that may harbor cryptic pockets, we applied our previously described fluctuation amplification of specific traits (FAST) simulation algorithm[42]. FAST is a goal-oriented adaptive sampling algorithm that exploits Markov state model (MSM) methods to explore regions of conformational space with user-specified structural features. An MSM is a network model of a protein's energy landscape which consists of a set of structural states the protein adopts and the rates of hopping between them[43,44]. After running FAST, we gathered additional statistics by running simulations from each state on the Folding@home distributed computing environment, which brings together the computing resources of hundreds of thousands of citizen scientists who volunteer to run simulations on their personal computers. Our final model has 4469 conformational states, providing a detailed characterization of the different structures the IID adopts, but making manual interpretation of the model difficult.

To identify cryptic pockets within the large ensemble captured by our MSM, we searched for signatures of cryptic pockets such as groups of residues with highly correlated changes in solvent exposure, referred to as exposons[12]. Exposons are often associated with cryptic sites because the opening/closing of such pockets

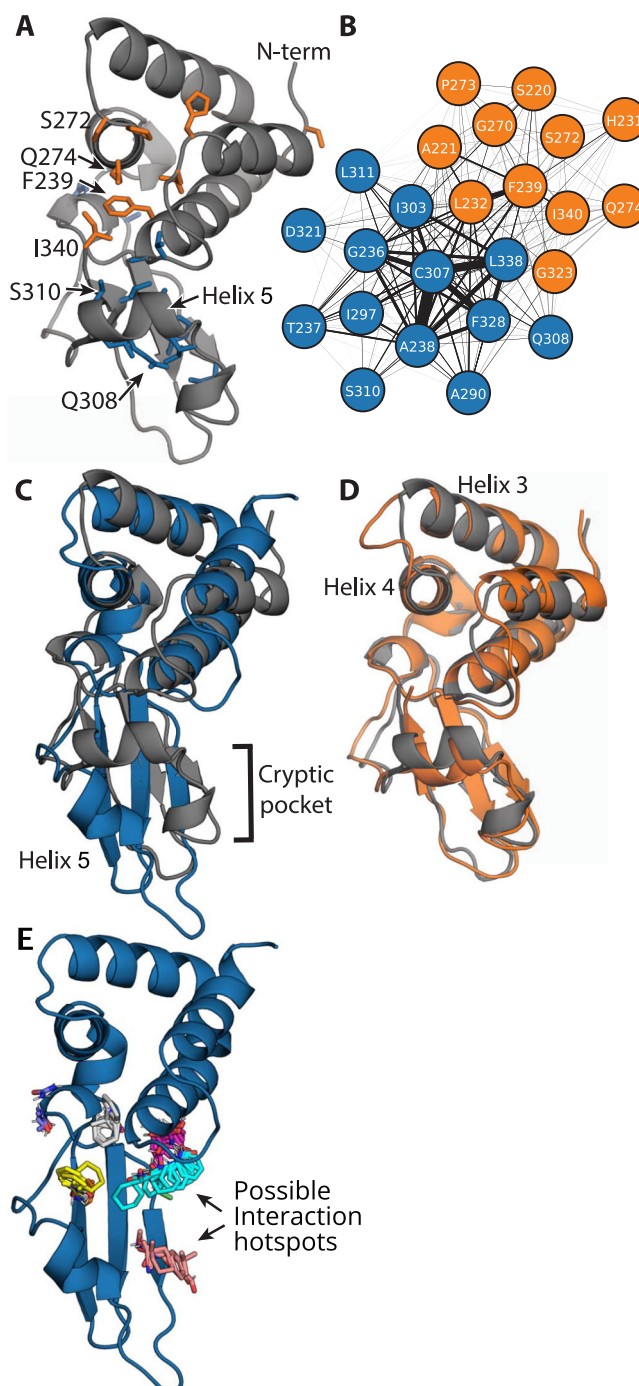

**Fig. 2 Exposons identify a large cryptic pocket and suggest potential allosteric coupling. A** Structure of VP35's IID highlighting residues in two exposons (blue and orange), the N-terminus (N-term), and C-terminus (I340) (PDB ID 3FKE). **B** Network representation of the coupling between the solvent exposure of residues in the two exposons. The edge width between residues is proportional to the mutual information between them. **C** Structure highlighting the opening of a cryptic pocket via the displacement of helix 5 that gives rise to the blue exposon. **D** Structure highlighting the conformational change that gives rise to the orange exposon overlaid on the crystal structure (gray) to highlight that the rearrangements are subtler than in the blue exposon. **E** FTMap results for the main cryptic pocket as shown in (**C**) and hotspots where a variety of small organic probes (multicolored sticks) form energetically favorable interactions. The probe molecules are intended to capture different drug-like interactions (such as hydrogen bonding and Van der Waals contacts) and include acetamide, acetonitrile, acetone, acetaldehyde, methylamine, benzaldehyde, benzene, isobutanol, cyclohexane, N,N-dimethylformamide, dimethyl ether, ethanol, ethane, phenol, isopropanol, or urea[47,68–70].

with the dsRNA backbone in the dsRNA-bound crystal structure[33], so targeting this cryptic pocket could directly disrupt this binding mode.

Retrospective analysis of other validated drug targets suggests cryptic sites created by the movement of secondary structure elements, such as the displacement of helix 5, are often druggable[45]. The potential druggability of this cryptic site is also supported by the application of the Fpocket and FTMap algorithms[46,47]. Fpocket predicts this cryptic site to have a high druggability score (0.681) and FTMap highlights a number of hotspots within the pocket where small molecules could form a variety of energetically favorable interactions (Fig. 2E and Supplementary Fig. 1). Unfortunately, disrupting backbone binding is of less therapeutic utility than disrupting blunt-end binding and it is unknown whether the contacts between A306, K309, and S310 are essential for backbone binding. Therefore, it is unclear from this analysis alone whether drugging this newly discovered cryptic pocket would be useful.

The second exposon (orange in Fig. 2) encompasses portions of both dsRNA-binding interfaces, but it does not correspond to a cryptic pocket. This cluster includes residues that bind dsRNA's backbone (i.e., S272) and residues that interact with both the blunt ends and backbone of dsRNA (i.e., F239, Q274, and I340)[33]. Therefore, altering the conformational preferences of the second exposon could potentially disrupt the blunt-end-binding mode and its crucial role in Ebola virus's ability to evade an immune response. However, the largest conformational change involved in the formation of this exposon is a displacement of the loop between helices 3 and 4 (Fig. 2D and Supplementary Movie 2). This rearrangement does not create a cryptic pocket that is large enough to accommodate drug-like molecules, so it is not obvious how to directly manipulate the orange exposon.

**The cryptic pocket is allosterically coupled to the blunt-end-binding interface.** Even though the cryptic pocket does not coincide with the interface of VP35's IID that binds dsRNA blunt ends, it could still serve as a cryptic allosteric site that allosterically controls dsRNA binding. Indeed, the physical proximity of the two exposons and the coupling between them both hint at the possibility for allosteric coupling. Furthermore, our exposons analysis could easily underestimate this coupling given that it focuses on correlated transitions of residues between solvent-exposed and completely buried states, leaving it blind to more subtle conformational fluctuations and allostery involving residues that are always buried (or always exposed).

gives rise to cooperative increases/decreases in the solvent exposure of surrounding residues. Importantly, once an exposon has been identified, our MSM framework provides a facile means to identify the conformational changes that give rise to that exposon.

Our simulations reveal two exposons in the VP35 IID, one of which corresponds to a large cryptic pocket. The blue exposon (Fig. 2A, B) which overlaps with the backbone-binding interface in Fig. 1, consists of a set of strongly coupled residues in helix 5 and adjacent loops and secondary structure elements. Visualizing the conformational change that gives rise to this cluster reveals a substantial displacement of helix 5, creating a large cryptic pocket between it and the helical domain (Fig. 2C and Supplementary Movie 1). A number of residues that are displaced along with helix 5 (i.e., A306, K309, and S310) make Van der Waals contacts

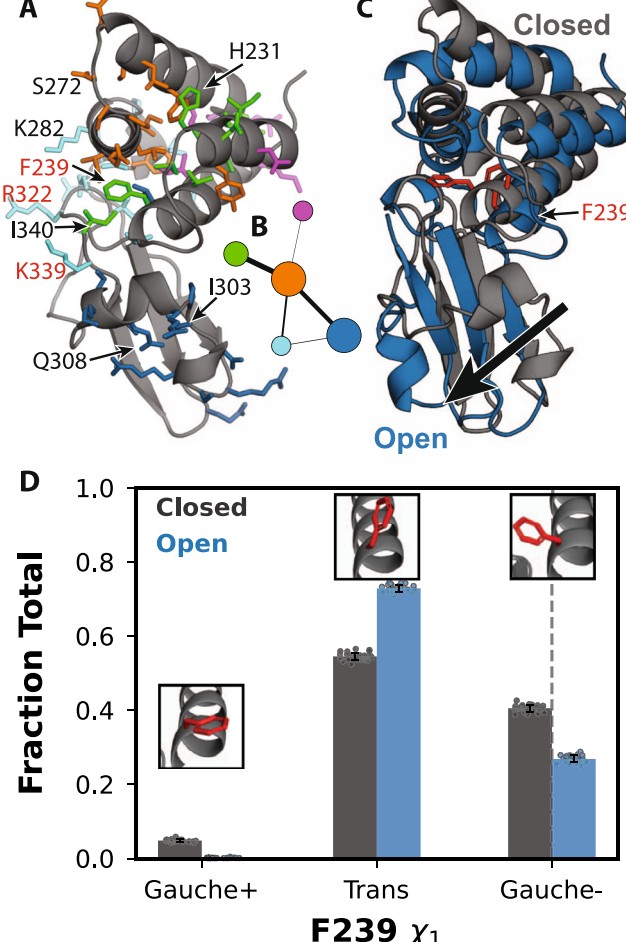

**Fig. 3 Allosteric network revealed by the CARDS algorithm. A** Structure of VP35's IID with residues in the allosteric network shown in sticks and colored according to which of five communities they belong to. Substitution of residues labeled in red with alanine disrupts binding to dsRNA blunt ends and results in a dramatic reduction in immune suppression. **B** Network representation of the coupling between communities of residues, colored as in (**A**). Node size is proportional to the strength of coupling between residues in the community, and edge widths are proportional to the strength of coupling between the communities. **C** Representative states of the correlated changes from the DiffNet. In gray is a structure with a closed pocket and in blue is a structure from MD simulation with an open pocket. F239 is shown in red sticks for orientation. **D** Distribution of F239 $\chi_1$ from the MSM with respect to states wherein the pocket is open (blue) or closed (black) for the three rotamers. The bar height is mean value from 25 bootstrapped MSMs (dots) of the sum of the population of all states in the MSMs with the specified rotamer. Insets show the conformation of F239 with the highest probability within the region of a given peak in the distribution as sampled in our MSM. The black dashed line at the Gauche position corresponds to the calculated value of F239 $\chi_1$ from PDB 3L26. Error bars are standard deviation from the mean of bootstrapped values from recalculating the MSM twenty-five times (see "Methods"). Source data are provided as a Source Data file.

To explore the potential for a broader allosteric network, we quantified the allosteric coupling between every pair of residues using our correlation of all rotameric and dynamical states (CARDS) algorithm[48]. CARDS classifies each dihedral in each snapshot of a simulation as being in one of three rotameric states (gauche+, gauche-, or trans) and one of two dynamical states (ordered or disordered). A mutual information metric is then used to quantify the coupling between the structure and dynamics of every pair of dihedral angles, which can then be coarse-grained to the correlation between every pair of residues. Importantly, CARDS accounts for the potential role of residues that are always buried or always exposed to solvent and subtle conformational changes that do not alter the solvent exposure of residues.

CARDS reveals a broader allosteric network than that identified by our exposons analysis and suggests strong coupling between the cryptic pocket and blunt-end-binding interface (Fig. 3A, B). This network consists of five communities of strongly coupled residues, four of which coincide with large portions of the two dsRNA-binding interfaces. One of these communities (orange) is a hub in the network, having significant coupling to all the other communities. It encompasses part of the orange exposon, particularly residues around the loop between helices 3 and 4. The orange CARDS community and exposon both capture Q274, which engages in both dsRNA-binding interfaces, and S272, which contacts the backbone[33]. However, the CARDS community includes many additional residues not captured by exposons analysis. Examples include I278, which engages in both dsRNA-binding interfaces, and D271, which is part of the PPI between the two binding modes[33]. One of the orange community's strongest allosteric connections is to the green community. This community encompasses the rest of the residues in the orange exposon, including F239 and I340, which are part of both dsRNA-binding interfaces[33]. The green community also captures additional residues, reaching deep into the helical domain. The orange community is also strongly coupled to the blue community, which includes much of helix 5 and nearby residues that move to give rise to the cryptic pocket that was captured by the blue exposon. Notably, the orange and blue communities are both coupled to a cyan cluster that was not hinted at by our exposons analysis because the residues involved are always solvent-exposed. It includes R322, which is part of the blunt-end-binding interface and the PPI between the two binding modes, and K282, which also contacts dsRNA blunt ends[33]. In addition, this community includes K339, which is an important determinant of the electrostatic favorability of dsRNA binding[33]. Together, these results suggest that opening of the cryptic pocket could strongly impact residues involved in both dsRNA-binding interfaces, as well as the PPI between the two binding modes.

**Opening of the cryptic pocket alters the structural preferences of the dsRNA-binding interface.** To assess if pocket opening impacts the blunt-end-binding interface, we compared the ensembles of structures with the cryptic pocket open or closed. We hypothesized that if pocket opening affects blunt-end binding, the dsRNA-binding residues in the ensembles of structures of the open and closed states will have distinct structural features other than pocket opening. To test this hypothesis, we applied our previously described machine learning algorithm, DiffNets, which is a supervised autoencoder architecture designed to identify the key differences between two or more structural ensembles[10]. In this case, we used DiffNets to compare the ensemble of structures with an open cryptic pocket to those with a closed cryptic pocket and assess if there are important differences between the structural preferences of the blunt-end-binding interface.

This analysis reveals significant coupling between the opening/closing of the cryptic pocket and the structural preferences of a key blunt-end-binding residue, F239. Specifically, we found that the distance between F239 and helix 5 is strongly correlated with the extent of the pocket opening. Further investigation revealed that the distribution of $\chi_1$ angles for F239 when the pocket is open differs substantially from the distribution when the pocket is closed (Fig. 3D). The orientation of F239 observed in available

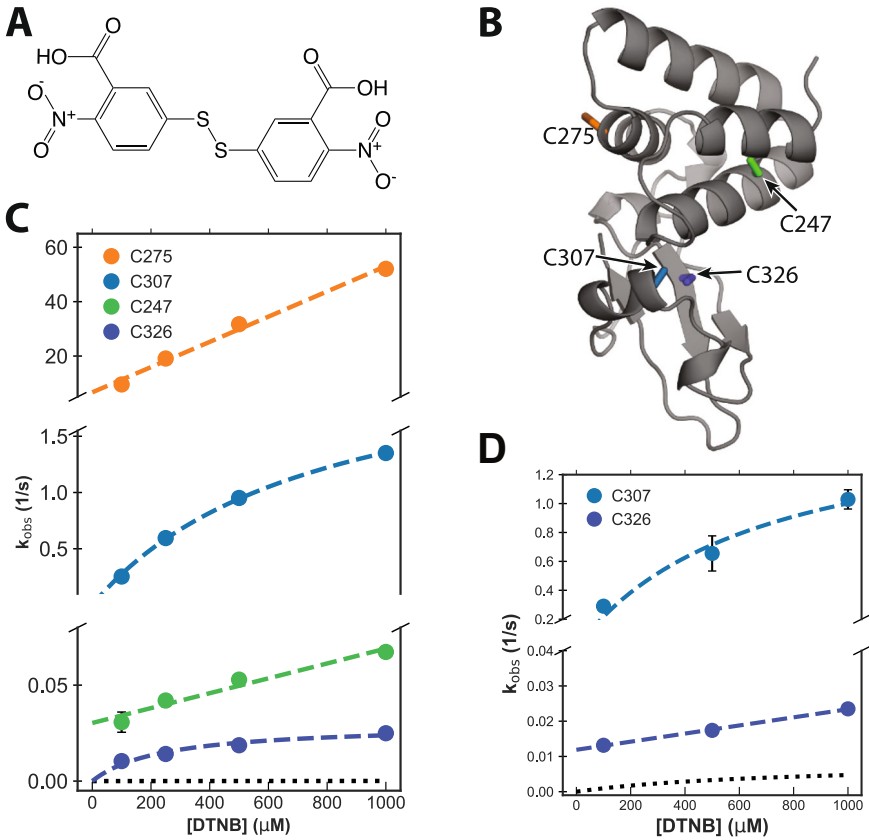

**Fig. 4 Thiol labeling supports the existence of the predicted cryptic pocket. A** Structure of the DTNB-labeling reagent. **B** Structure of VP35's IID highlighting the locations of the four native cysteines (sticks). C307 and C326 are both buried and point into the predicted cryptic pocket. **C** Observed labeling rates (circles) for WT VP35 at a range of DTNB concentrations. Fits to the Linderstrøm–Lang model are shown in dashed colored lines and the expected labeling rate from the unfolded state is shown as black dotted lines. The mean and standard deviation from three replicates is shown but error bars are generally smaller than the symbols. **D** Observed labeling rates (circles) for VP35 C247S/C275S. Fits to the Linderstrøm–Lang model are shown in dashed colored lines, and the expected labeling rate from the unfolded state is shown as black dotted lines. The mean and standard deviation from three replicates is shown but error bars are generally smaller than the symbols. Source data are provided as a Source Data file.

crystal structures is a well-populated when the cryptic pocket is closed. Opening of the cryptic pocket is associated with a reduction in the probability of this Gauche- dsRNA-binding competent rotamer. Therefore, we propose that stabilizing the closed pocket should enhance the affinity between VP35 and dsRNA blunt ends, while stabilizing the open pocket (e.g., via binding of a small molecule) should disrupt dsRNA binding.

**Thiol labeling experiments corroborate the predicted cryptic pocket**. One way to experimentally test our prediction of a cryptic pocket is to probe for solvent exposure of residues that are buried in all the structures that are currently available in the protein data bank (PDB) but become exposed to solvent upon pocket opening. Cysteines are particularly appealing candidates for such experiments because (1) they have a low abundance and (2) their thiol groups are highly reactive, so it is straightforward to detect exposed cysteines by introducing labeling reagents that covalently bind accessible thiols. Fortuitously, VP35's IID has two cysteines (C307 and C326) that are buried in available crystal structures but become exposed to solvent when the cryptic pocket opens (Fig. 4B). There is also a cysteine (C275) that is on the surface of the apo crystal structure[32] and a fourth cysteine (C247) that is buried in the helical bundle. C275 is typically solvent-exposed in our simulations, as expected based on the crystallographic data. Examining the solvent exposure of C247 revealed it is sometimes exposed to solvent via an opening of helix 1 relative to the rest of the helical bundle (Supplementary Fig. 2), but FTMap did not

identify any hotspots that are likely to bind drug-like molecules in this region. Therefore, we expect to observe labeling of all four cysteines on a timescale that is faster than global unfolding of the protein.

To experimentally test our predicted pocket, we applied a thiol labeling technique that probes the solvent exposure of cysteine residues[49]. For these experiments, 5,5'-dithiobis-(2-nitrobenzoic Acid) (also known as DTNB or Ellman's reagent, Fig. 4A) is added to a protein sample. Upon reaction with the thiol group of an exposed cysteine, DTNB breaks into two TNB molecules, one of which remains covalently bound to the cysteine while the other is released into solution. The accumulation of free TNB can be quantified based on the increased absorbance at 412 nm. We have previously applied this technique to test predicted pockets in β-lactamase enzymes[12,50].

As expected from our computational model, the observed signal from our thiol labeling experiments is consistent with opening of the cryptic pocket (Fig. 4C). Absorbance curves are best fit by four exponentials, each with an approximately equivalent amplitude that is consistent with expectations based on the extinction coefficient for DTNB (Supplementary Fig. 3). To assign these labeling rates to individual cysteines, we systematically mutated the cysteines to serines, performed thiol labeling experiments, and assessed which rates disappeared and which remained (Supplementary Fig. 4 and Table 1). For example, labeling of the C275S variant lacks the very fastest rate for wild-type, consistent with the intuition that a residue that is

surfaced exposed in the crystal structure (i.e., C275) should label faster than residues that are generally buried. The consistency of the labeling rates between variants also confirms none of the observed labeling events are dependent on labeling of other cysteine residues.

To test whether the observed labeling could be due to an alternative process, such as global unfolding, we determined the population of the unfolded state and unfolding rate of VP35's IID under native conditions (Supplementary Table 2) and the intrinsic labeling rate for each cysteine (Supplementary Table 3). As shown in Fig. 4C, the observed labeling rates are all considerably faster than the expected labeling rate from the unfolded state at a range of DTNB concentrations. This result confirms that labeling of all four cysteines arises from fluctuations within the native state, consistent with our computational predictions.

That all four cysteines undergo labeling suggests that C247 undergoes local fluctuations that our exposons analysis does not predict will form a pocket. To determine the importance of this fluctuation, we calculated the equilibrium constant for the exposure of both C247 and C307. Opening of the cryptic pocket is far more probable than the structural fluctuation that exposes C247 (equilibrium constants for the exposure of C247 and C307 are $6.9 \times 10^{-4} \pm 7.0 \times 10^{-5}$ and $4.0 \times 10^{-1} \pm 1.0 \times 10^{-2}$, respectively). Therefore, a ligand would have to pay a greater energetic cost to stabilize the conformational change that exposes C247 than to stabilize the open state of the cryptic allosteric site created by the motion of helix 5. Taken together with the fact that the motion of helix 5 creates a more druggable pocket than the motion that exposes C247, we continue to focus on the cryptic pocket created by the helix 5 motion.

**Mutations support our predicted allosteric network**. We sought to test our model of allostery in VP35 by introducing mutations and assessing their impact on the conformations of distal sites. To select mutations, we drew on both our model of allosteric coupling and the published literature. For example, our model's prediction of coupling between the conformation of F239 (in the blunt-end-binding interface) and cryptic pocket opening suggests that an F239A mutation is likely to alter pocket opening. Previous work suggests that the linker between the two domains of Reston VP35 confers it with greater stability and rigidity than the Zaire variant of VP35 we focus on in this work[51]. One of the significant differences between the linkers of the two proteins is the presence of a proline in Reston VP35. Given proline is conformationally restricted, we reasoned that substituting A291 for proline in the linker of Zaire VP35 may restrict cryptic pocket opening and enhance dsRNA binding. To test these predictions, we created the relevant variants of VP35 and measured their impact on cryptic pocket opening using our thiol labeling assay.

Thiol labeling of F239A demonstrates that the mutation allosterically increases opening of the cryptic pocket. We find that the observed labeling rates for the cysteines in the cryptic pocket are twofold faster than in wild-type VP35. Fitting with the Linderstrøm–Lang model reveals that the equilibrium probability of C307 exposure in F239A is approximately double that of wild-type ($1.1 \pm 0.2$ vs $4.0 \times 10^{-1} \pm 1.0 \times 10^{-2}$, respectively) (Supplementary Fig. 5). These thiol labeling data suggest that communication flows to and from the end-cap involved dsRNA binding residues and cryptic pocket.

In contrast, the mutation A291P decreases the probability of pocket opening, which results in a higher affinity for dsRNA. Thiol labeling experiments reveal that A291P dramatically reduces the labeling rates of the two cysteines in the cryptic pocket (Supplementary Fig. 6). In fact, the labeling rate of C326 in

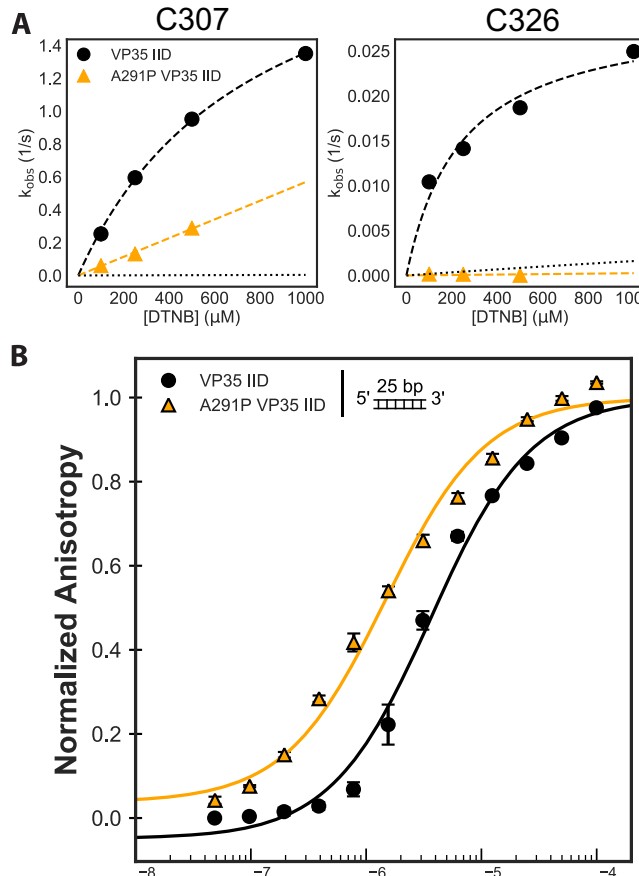

**Fig. 5 An A291P mutation favors both the closed cryptic pocket in VP35's IID and increases dsRNA binding. A** DTNB observed labeling rates of both wild-type and the A291P mutation for the two cysteines in the pocket. All four cysteines are present, complete data for all four observed rates are in Supplemental Fig. 4. **B** Binding of both C247S/C275S and A291P VP35 IID to a fluorescently labeled 25-bp double-stranded RNA. The anisotropy was calculated from measured fluorescence polarization and fit to a single-site binding model (black and orange lines). The means and standard deviations from three replicates are shown but error bars are generally smaller than the symbols. Anisotropy was normalized to the max anisotropy for each dataset. Source data are provided as a Source Data file.

the A291P background is similar to the rate of global protein unfolding (Fig. 5A), suggesting that the pocket never opens enough to expose the most deeply buried regions of the cryptic pocket observed in the wild-type protein. The probability that the C307 of the A291P variant is accessible to our DTNB-labeling reagent is also significantly smaller than in wild-type ($1.4 \times 10^{-4} \pm 2.0 \times 10^{-4}$ vs $4.0 \times 10^{-1} \pm 1.0 \times 10^{-2}$, respectively).

**Stabilizing the closed pocket increases dsRNA binding**. Based on our predicted allosteric network, stabilizing the closed state of the cryptic pocket should enhance dsRNA binding. Specifically, the fact that the pocket is closed in the co-crystal structure of VP35 with dsRNA (PDB 3L26) implies a closed pocket is favorable for dsRNA binding and a mutation that stabilizes the pocket in its closed form would increase dsRNA binding. Therefore, we should see a higher affinity between A291P and dsRNA.

To test this prediction, we developed a fluorescence polarization (FP) assay for measuring the affinity of VP35 for

dsRNA. Paralleling past work on VP35-peptide interactions[40], we added varying concentrations of VP35 IID to a fixed concentration of 25-bp RNA with a fluorescein isothiocyanate (FITC) conjugation at the 5' end (Supplementary Table 4). Free FITC-dsRNA emits depolarized light upon excitation with polarized light because of the molecule's fast rotation. Binding of one or more VP35 molecules restricts the motion of FITC-dsRNA, resulting in greater emission of polarized light, which is best monitored by the change in anisotropy[52]. This anisotropy-based binding measurement recapitulates previously published binding affinities for two different dsRNA end topologies (blunt or overhanging 3' ends) (Supplementary Fig. 8).

Our data show that closing the pocket with A291P increases dsRNA binding. To test how A291P binds dsRNA, we repeated the binding assay done with VP35 IID C247S/C275S with A291P and a 25 base-pair blunt-ended dsRNA and calculated the apparent affinity to be $1.8 \pm 0.1\,\mu M$. This corresponds to a twofold increase in apparent binding affinity relative to wild-type VP35. We also find that A291P is sensitive to the presence of a 3' overhang as characterized by a rightward shift of the binding curve (Supplementary Fig. 9).

**Stabilizing the open cryptic pocket allosterically disrupts binding to dsRNA blunt ends**. We reasoned that covalent attachment of TNB to the cysteine sidechains pointing into the pocket (C307 and C326) would provide a means to capture the open pocket and assess the impact of stabilizing this state with a drug-sized probe on dsRNA binding. The addition of TNB to these cysteines is sterically incompatible with the closed conformation of VP35's RNA-bound IID that has been observed crystallographically. TNB's mass of ~198 Da is also similar to many drug fragments used in screening campaigns, making it a reasonable surrogate for the type of effect one might achieve with a fragment hit. Given that we already know DTNB labels the IID's cysteines, a TNB-labeled sample is easily obtainable by waiting until the labeling reaction goes to completion. Finally, we have previously used this same strategy to identify cryptic pockets that exert allosteric control over the activity of β-lactamase enzymes[12,50].

To specifically probe the behavior of effects of labeling the cryptic pocket, we focus on a C247S/C275S variant that only has cysteines in the cryptic pocket. As with the wild-type protein, thiol labeling of the C247S/C275S variant is consistent with the formation of the predicted cryptic pocket (Fig. 4D).

Comparing the dsRNA-binding profile of TNB-labeled protein (TNB-VP35 IID) to unlabeled protein reveals that labeling allosterically reduces the affinity for blunt-ended dsRNA by at least fivefold (Fig. 6A). Solubility limitations prevented us from observing complete binding curves for labeled protein, but the data are sufficient to demonstrate that TNB-labeling has at least as strong an effect on binding as addition of a 3' overhang. As a control to ensure that labeling does not disrupt binding by simply unfolding the protein, we measured the circular dichroism (CD) spectra of labeled and unlabeled protein. The similarity between the CD spectra (Fig. 6B) demonstrates that the IID's overall fold is not grossly perturbed. Previous work demonstrated that VP35's two subdomains do not fold independently[32] supporting our proposal that both domains remain mostly folded. These data indicate that the change in dsRNA binding from TNB-labeled VP35 is unlikely to be due to a local unfolding of the β-sheet subdomain. Furthermore, since past work demonstrated that reducing the blunt-end-binding affinity by as little as threefold is sufficient to allow a host to mount an effective immune response[33,34], targeting our cryptic pocket could be of great therapeutic value.

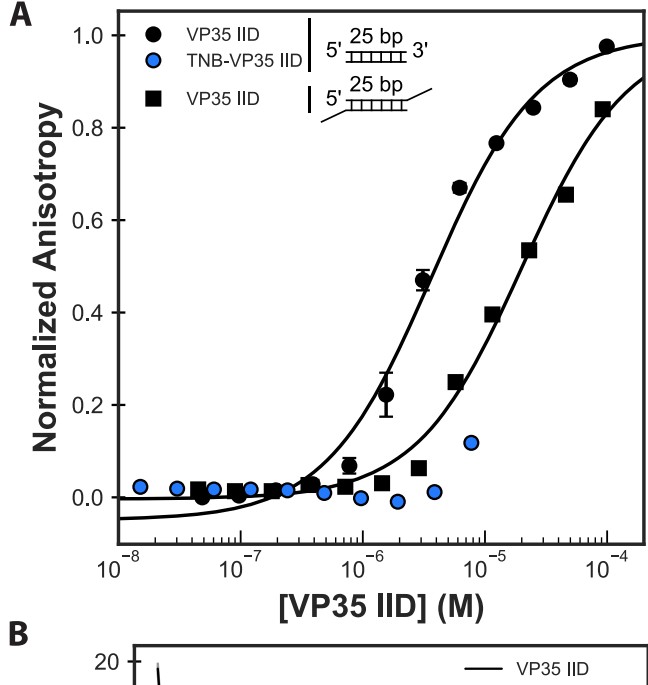

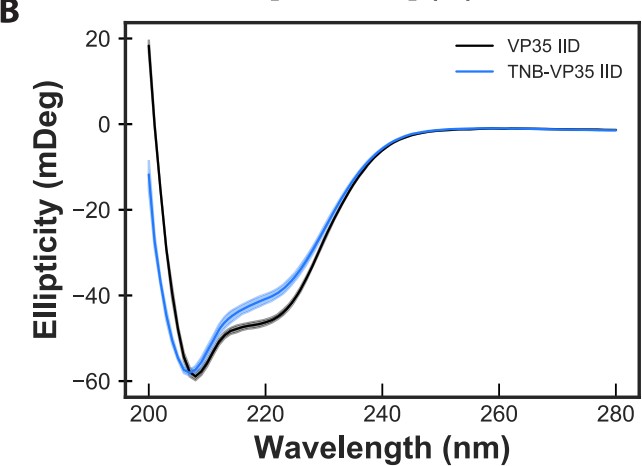

**Fig. 6 Stabilizing the open cryptic pocket in VP35's IID disrupts dsRNA binding. A** Binding of unlabeled C247S/C275S VP35 IID to two different dsRNA constructs compared to binding of TNB-labeled protein to blunt-ended RNA. The two RNA constructs both have a 25-bp double-stranded segment, and one has 2 nucleotide overhangs on the 3' ends. The anisotropy was measured via a fluorescence polarization assay, converted to anisotropy, fit to a single-site binding model (black lines), and normalized to the fit maximum anisotropy. The mean and standard deviation from three replicates is shown but error bars are generally smaller than the symbols. **B** Circular dichroism (CD) spectra of labeled and unlabeled protein demonstrate that labeling does not unfold the protein. The opaque and semi-transparent lines represent the mean and standard deviation, respectively, from three replicates. CD spectra were collected in 50 mM sodium phosphate pH 7 at 50 μg/mL protein. Source data are provided as a Source Data file.

## Discussion

We have identified a cryptic allosteric site in the IID of the Ebola virus VP35 protein that provides a new opportunity to target this essential viral component. Past work identified several sites within the VP35 IID that are critical for immune evasion and viral replication[28,31,36,37], but structural snapshots captured crystallographically lacked druggable pockets[32,33]. We used adaptive sampling simulations to access more of the ensemble of conformations that VP35 adopts, uncovering an unanticipated cryptic pocket. While the pocket directly coincides with the

interface that binds the backbone of dsRNA, it was not clearly of therapeutic relevance since binding dsRNA's blunt ends is more important for Ebola's immune evasion mechanism[34]. However, our simulations also suggested the cryptic pocket is allosterically coupled to the blunt-end-binding interface and, therefore, could modulate this biologically-important interaction. Analysis of our computational model suggested that structures with an open cryptic pocket should be less compatible with binding to RNA blunt ends than structures with a closed pocket. Subsequent thiol labeling experiments confirmed that fluctuations within the folded state of the IID expose two buried cysteines that line the proposed cryptic pocket to solvent. Introducing an F239A mutation within the blunt-end-binding interface allosterically increases the probability of cryptic pocket opening[33,53,54]. An A291P mutation allosterically suppresses pocket opening and simultaneously increases the affinity of VP35 for dsRNA. Finally, covalently modifying the pocket facing cysteines to stabilize the open form of the cryptic pocket allosterically disrupts binding to dsRNA blunt ends by at least fivefold. Previous work demonstrated that reducing the binding affinity by as little as 3-fold is sufficient to allow a host to mount an effective immune response[33]. Therefore, it may be possible to attenuate the impact of viral replication and restrict pathogenicity by designing small molecules to target the cryptic allosteric site we report here.

More generally, our results speak to the power of simulations to provide simultaneous access to both hidden conformations and dynamics with atomic resolution. Such information is extremely difficult to obtain from single structural snapshots or powerful techniques that report on dynamics without directly yielding structures, such as NMR and hydrogen deuterium exchange. As a result, simulations are a powerful means to uncover unanticipated features of proteins' conformational ensembles, such as cryptic pockets and allostery, providing a foundation for the design of further experiments. We anticipate such simulations will enable the discovery of cryptic pockets and cryptic allosteric sites in other proteins, particularly those that are currently considered difficult targets. Furthermore, the detailed structural insight from simulations will facilitate the design of small-molecule drugs that target these sites.

## Methods

**Molecular dynamics simulations and analysis.** Simulations were initiated from the apoprotein model of PDB 3FKE[32,33] and run with Gromacs[55] using the amber03 force field[56] and TIP3P explicit solvent[57] at a temperature of 300 K and 1 bar pressure, as described previously[58]. Recombinant VP35 IID is known to be monomeric supporting our choice in system setup. We first applied our FAST-pockets algorithm[42] to balance (1) preferentially simulating structures with large pocket volumes that may harbor cryptic pockets with (2) broad exploration of conformational space. For FAST, we performed ten rounds of simulations with 10 simulations/round and 80 ns/simulation. To acquire better statistics across the landscape, we performed an RMSD-based clustering using a hybrid k-centers/k-medoids algorithm[59] implemented in Enspara[60] to divide the data into 1000 clusters. Then we ran three simulations initiated from each cluster center on the Folding@home distributed computing environment, resulting in an aggregate simulation time of 122 μs.

Exposons were identified using our previously described protocols[12], as implemented in Enspara[60]. Briefly, the solvent-accessible surface area (SASA) of each residue's side-chain was calculated using the Shrake-Rupley algorithm[61] implemented in MDTraj[62] using a drug-sized probe (2.8 Å sphere). Conformations were clustered based on the SASA of each residue using a hybrid k-centers/k-medoids algorithm, using a 2.7 Å$^2$ distance cutoff and five rounds of k-medoids updates. A Markov time of 6 ns was selected based on the implied timescales test (Supplementary Fig. 10). The center of each cluster was taken as an exemplar of that conformational state, and residues were classified as exposed if their SASA exceeded 2.0 Å$^2$ and buried otherwise. The mutual information between the burial/exposure of each pair of residues was then calculated based on the MSM (i.e., treating the centers as samples and weighting them by the equilibrium probability of the state they represent). Finally, exposons were identified by clustering the matrix of pairwise mutual information values using affinity propagation[63].

The CARDS algorithm[48] was applied to identify allosteric coupling using our established protocols[64], as implemented in Enspara[60]. Briefly, each dihedral angle

in each snapshot of the simulations was assigned to one of three rotameric states (gauche+, gauche-, or trans) and one of two dynamical states (ordered or disordered). The total coupling between each pair of dihedrals $X$ and $Y$ was then calculated as $I(X_R, Y_R) + I(X_R, Y_D) + I(X_D, Y_R) + I(X_D, Y_D)$, where $I$ is the mutual information metric, $X_R$ is the rotameric state of dihedral $X$, and $X_D$ is the dynamical state of dihedral $X$. The term $I(X_R, Y_R)$ is the purely structural coupling, while the sum of the other three terms is referred to as the disorder-mediated coupling. The dihedral level couplings were coarse-grained into residue-level coupling by summing the total coupling between all the relevant dihedrals. The network was subsequently filtered to only retain significant edges[65]. Finally, communities of coupled residues were identified by clustering the residue-level matrix of total couplings using affinity propagation[63]. These algorithms are available at github.com/bowman-lab.

We processed, trained, and analyzed our DiffNet as previously described[10]. Briefly, we isolated coordinates of the heavy atoms (all protein atoms excluding hydrogens) for trajectories of our two ensembles of pocket open and closed states using a 1.5 Å cutoff for the distance between the center of mass of residues 305–310 (helix 5) and the center of mass of residues 238 to 245 (helix 2). We then centered the atom coordinates at the origin and aligned to 3FKE. Next, we mean shifted then whitened the coordinates. Finally, we used the resulting data to train the neural net for 20 epochs with 30 latent space variables with a batch size of 32. Frames with the pocket closed were initially assigned a classification label of zero while frames with the pocket open were assigned a label of one. For expectation maximization, we set the initial bounds as 10–40% for closed frames then 60–90% for open frames. We then visualized the top 250 correlated distance changes in PyMol.

We used the calc_chi1 function in MdTraj to calculate the F239 $\chi_1$ in our original MSM. We then binned the $\chi_1$ values as guache+, trans, or gauche- using previously described cutoffs[66]. To estimate the error in our rotamer distribution in Fig. 3D, we randomly selected N trajectories from the original dataset where $N$ = number of total original trajectories in our dataset, with replacement. We then refit the MSM as described above, keeping the same state space but with the resampled trajectories, twenty-five total times. Then we calculated the total population of each rotamer in the resampled datasets, and the mean population of that rotamer across all resampled MSMs. The error bars are then the standard deviation of the mean of the resampled population values for each rotamer in the open and closed ensembles with respect to the refit MSMs.

**Protein expression and purification.** All variants of VP35's IID were purified from the cytoplasm of *E. coli* BL21(DE3) Gold cells (Agilent Technologies)[32–34,53,54]. Variants were generated using the site-directed mutagenesis method and confirmed by DNA sequencing. Transformed cells were grown at 37 °C until OD 0.3 then grown at 18 °C until induction at OD 0.6 with 1 mM IPTG (Gold Biotechnology, Olivette, MO). Cells were grown for 15 h then centrifuged after which the pellet was resuspended in 20 mM sodium phosphate pH 8, 1 M sodium chloride, with 5.1 mM β-mercaptoethanol. Resuspended cells were subjected to sonication at 4 °C followed by centrifugation. The supernatant was then subjected to Ni-NTA affinity (BioRad Bio-Scale Mini Nuvia IMAC column), TEV digestion, cation exchange (BioRad UNOsphere Rapid S column), and size-exclusion chromatography (BioRad Enrich SEC 70 column or Cytiva HiLoad 16/600 Superdex 75) into 10 mM HEPES pH 7, 150 mM NaCl, 1 mM MgCl$_2$, 2 mM TCEP.

**Thiol labeling.** We monitored the change in absorbance over time of 5,5'-dithiobis-(2-nitrobenzoic acid) (DTNB, Ellman's reagent, Thermo Fisher Scientific). Various concentrations of DTNB were added to the protein, and change in absorbance was measured in either an SX-20 Stopped Flow instrument (Applied Photophysics, Leatherhead, UK), or an Agilent Cary60 UV–vis spectrophotometer at 412 nm until the reaction reached a steady state (~300 s). Data were fit with a Linderstrøm–Lang model to extract the thermodynamics and/or kinetics of pocket opening, as described in detail previously[12]. As a control, the equilibrium constant for folding and the unfolding rate were measured (Supplementary Table 2) and used to predict the expected labeling rate from the unfolded state. The equilibrium constant was inferred from a two-state fit to urea melts monitored by fluorescence and unfolding rates were inferred from exponential fits to unfolding curves monitored by fluorescence after the addition of urea, as described previously[12,50,67]. Fluorescence data were collected using a Jasco FP-8300 Spectrofluorometer with Jasco ETC-815 Peltier and Koolance Exos2 Liquid Coolant-controlled cuvette holder.

**Fluorescence polarization binding assay.** Apparent binding affinities between variants of VP35's IID and dsRNA were measured using fluorescence polarization in 10 mM Hepes pH 7, 150 mM NaCl, 1 mM MgCl$_2$. A 25 base pair FITC-dsRNA (Integrated DNA Technologies) substrate with and without a 2 nucleotide 3' overhang was included at 100 nM. The sample was equilibrated for one hour before data collection. Data were collected on a BioTek Synergy2 Multi-Mode Reader as polarization and were converted to anisotropy as described previously[52]. TNB-labeled samples were generated by allowing DTNB and VP35's IID to react for 3 min and then removing excess DTNB with a Zeba spin desalting columns (Thermo Fisher Scientific). Data were analyzed in Jupyter Notebook using Scipy 1.3.2, NumPy 1.14.x and 1.19.5, Matplotlib 3.5, Pandas 0.25.3, and Seaborn 0.11.2.

A single-site binding model was sufficient to fit the data:

$$r_{obs} = r_0 + (r_{max} - r_0) * \left( \frac{K_A * [VP35]}{1 + K_A * [VP35]} \right) \qquad (1)$$

**Reporting summary**. Further information on research design is available in the Nature Research Reporting Summary linked to this article.

## Data availability

The data and molecular dynamics datasets that support this study are available from the corresponding author upon reasonable request. MD start files are available with the MSM data in the below linked repository. The MSM data and MD starting structures have been deposited in the Open Science Framework database https://osf.io/5pg2a. Referenced structures are: PDB ID 3FKE and PDB 3L26. Source data are provided with this paper.

## Code availability

FAST, Enspara (including Exposons and CARDS), and DiffNets are freely available software packages on GitHub at https://github.com/bowman-lab/diffnets. Jupyter Notebooks used to analyze experimental data are available upon request.

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

## Acknowledgements

We are grateful to the citizen scientists who participate in Folding@home for volunteering to run simulations on their personal computers. This work was funded by NSF CAREER Award MCB-1552471 and NIH grant R01 GM124007 (Bowman), as well as NIH grants R01AI123926, P01AI120943, and R01AI143292 (Amarasinghe). G.R.B. holds a Career Award at the Scientific Interface from the Burroughs Wellcome Fund and a Packard Fellowship for Science and Engineering from The David & Lucile Packard Foundation. M.A.C. was supported by the NIH grants 5R25GM103757 to WUSTL IMSD program, and NIH F31AI157079. S.S. was supported by a MilliporeSigma Fellowship. We thank Drs. Timothy M. Lohman and Alexander G. Kozlov for advice on FP assays.

## Author contributions

M.A.C., T.E.F., U.L.M., and K.E.M. conducted experiments and analyzed the data. M.A.C., U.L.M., N.V., and M.I.Z. performed simulations and M.A.C., T.E.F., U.M., S.S., N.V., M.I.Z., and J.R.P. analyzed simulation data. M.A.C., G.R.B., and G.K.A. acquired funding. M.A.C., G.R.B., G.K.A. conceptualized the research direction and strategy. M.A.C., T.E.F., G.R.B., and G.K.A. contributed to manuscript writing and all authors contributed to editing. M.A.C., G.R.B., and T.E.F. created data visualizations.

## Competing interests

The authors declare no competing interests.
