## [Peer Review File · Nature Communications]

A cryptic pocket in Ebola VP35 allosterically controls RNA bindingReviewers' Comments:

Reviewer #1:

Remarks to the Author:

In the current manuscript, the authors expected to discover cryptic allosteric sites in Ebola viral protein (VP35) using molecular dynamics simulations and machine learning methods and designed biochemical experiments to support the predicted cryptic allosteric sites. Considering the importance of cryptic allosteric sites in drug design and the challenge in the identification of cryptic allosteric sites, I think that the research presented by Cruz et al. is very interesting and it fits into the scope of Nature Communications. However, before the publication, some of the major concerns should be addressed:

(1) My major concern is about whether the predicted cryptic allosteric site is a site for drug-molecule binding. In general, a drug binding site is formed by several residues that contribute to bind drug molecules. However, in the current manuscript, the authors only focused on the two cysteine residues (C307 and C326). The biochemical data based on only two residues are not enough to validate the authors' hypothesis that they formed a cryptic allosteric site. Thus, I suggest authors to execute additional computations and experiments to strength their hypothesis. Recent papers have reported a promising strategy to identify cryptic allosteric sites using combined computational and experimental methods (PMID: 34354057, 34163609; DOI: 10.1016/j.apsb.2021.06.015; DOI: 10.1021/acscatal.9b02556).

Authors should first use the pocket detection program such as FPOCKET to predict cryptic allosteric sites in the representative conformation of VP35 and identify residues constituting the predicted cryptic allosteric sites. Then, mutations of all residues in the predicted cryptic allosteric site to reveal whether mutations had effects on the interactions of VP35-dsRNA. (2) Identification of a cryptic allosteric site: How can this be defined a binding "site"? The statement that the authors discovered a potential allosteric site appearing weak, without an actual proof to target this site with a small molecule. At least a docking analysis could be provided in the supporting information.

(3) My second concern is about the simulated systems. Authors used the apo VP35 protein (PDB ID: 3FKE) in their MD simulations. This PDB structure of VP35 is a homodimer without dsRNA. However, in the crystal structure of VP35-dsRNA complex (PDB ID: 3L25), the structure of VP35 is a homotetramer. The authors should claim the VP35 homodimer physiologically existed in virus. If the VP35 protein is in the homotetramer form in the physiological state, the construction of the simulated systems in the current study was artificial. Furthermore, I suggest authors using PDB 3L25 to perform additional simulations, in which dsRNA was removed from the structural complex to simulate the homotetramer. Authors should validate whether the predicted cryptic allosteric site in the homodimer protein reproduced in the homotetramer protein.

(4) The abstract and discussion sections regarding to the method development in the identification of cryptic allosteric sites are insufficient. Recent work from other groups should be added.

(5) Movie S1 and Movie S2: The labels of key domains would facilitate the reader to interpret the movies. One could also think about some arrows that would further suggest the region of protein on which the reader should focus, according to Authors interpretation. In addition, in Figure 1, helix number should be labeled that would facilitate the reader to understand authors' structural description in the text.

Reviewer #2:

Remarks to the Author:

Disrupting protein-nucleic acid interactions (PNIs) with small molecules is a promising therapeutic strategy. Unfortunately, most known PNIs lack evident druggable pockets and are thus considered intractable targets.

Cryptic (hidden) pockets might provide a valid alternative approach for designing drugs targeting PNIs. However, by their nature cryptic pockets are not visible in the unliganded (apo) crystal structure of the target proteins. Methods based on enhanced sampling molecular dynamics simulations can help in this respect. In this paper the authors show how a combination of adaptive sampling molecular

dynamics simulations with thiol labeling and mutagenesis experiments can be used to successfully identify a cryptic allosteric pocket allosterically coupled to a protein-nucleic acid interface.

The chosen protein, the interferon inhibitory domain (IID) of Ebola viral protein 35 (VP35), plays a crucial role in antagonizing the host's innate immunity and thus is a promising drug target. However, crystal structures have shown that the binding interfaces of VP35 IID with dsRNA are flat and lack evident cavities. This makes it difficult to design small molecules that bind at the interface and disrupt the PNIs with classic computer aided approaches.

In contrast to classic (static approaches) adaptive sampling simulations and machine learning algorithms are able predict a cryptic pocket that is allosterically coupled to a key dsRNA-binding interface. By performing thiol labeling and mutagenesis experiments the authors are able to convincingly validate the predicted pocket, while covalent modifications that mimic drug binding at the cryptic pocket successfully disrupt dsRNA binding.

The successful identification of a cryptic pocket allosterically coupled to a PNIs shows the power of predictive protein dynamics simulations in the rational targeting of 'intractable' proteins and paves the avenue for the use of combined computational and experimental pipelines to target other difficult PNIs.

Response to Reviewer #1:

Reviewer #1 (Remarks to the Author):

In the current manuscript, the authors expected to discover cryptic allosteric sites in Ebola viral protein (VP35) using molecular dynamics simulations and machine learning methods and designed biochemical experiments to support the predicted cryptic allosteric sites. Considering the importance of cryptic allosteric sites in drug design and the challenge in the identification of cryptic allosteric sites, I think that the research presented by Cruz et al. is very interesting and it fits into the scope of Nature Communications. However, before the publication, some of the major concerns should be addressed:

(1) My major concern is about whether the predicted cryptic allosteric site is a site for drug-molecule binding. In general, a drug binding site is formed by several residues that contribute to bind drug molecules. However, in the current manuscript, the authors only focused on the two cysteine residues (C307 and C326). The biochemical data based on only two residues are not enough to validate the authors' hypothesis that they formed a cryptic allosteric site. Thus, I suggest authors to execute additional computations and experiments to strength their hypothesis. Recent papers have reported a promising strategy to identify cryptic allosteric sites using combined computational and experimental methods (PMID: 34354057, 34163609; DOI: 10.1016/j.apsb.2021.06.015; DOI: 10.1021/acscatal.9b02556). Authors should first use the pocket detection program such as FPOCKET to predict cryptic allosteric sites in the representative conformation of VP35 and identify residues constituting the predicted cryptic allosteric sites. Then, mutations of all residues in the predicted cryptic allosteric site to reveal whether mutations had effects on the interactions of VP35-dsRNA.

Response: We thank the reviewer for their suggestions. To better demonstrate the unique capacity of the open pocket state to bind druglike molecules as compared to the closed state found in the crystal structures, we applied Fpocket to PDB model 3FKE and the program did not identify a pocket where we found one using simulation. Additionally, when we apply Fpocket to the most open, most populated state sampled in our MD data, Fpocket supports our reasoning. The open pocket is large enough to accommodate drug sized molecules and has a high (> 0.5) druggability score.

- We added a mention of our application of Fpocket to the fourth paragraph of the results section line 38 (underlined text): Retrospective analysis of other validated drug targets suggests cryptic sites created by the movement of secondary structure elements, such as the displacement of helix 5, are often druggable.⁴⁶ The potential druggability of this cryptic site is also supported by application of the Fpocket and FTMap algorithms.^{47,48} Fpocket predicts this cryptic site to have a high druggability score (0.681) and FTMap highlights a number of hotspots within the pocket where small molecules could form a variety of energetically-favorable interactions (Fig. 2E and Supplementary Fig. 1). Unfortunately, disrupting backbone binding is of less therapeutic utility than disrupting blunt end binding and it is unknown whether the contacts between A306, K309, and S310 are essential for backbone binding. Therefore, it is unclear from this analysis alone whether drugging this newly discovered cryptic pocket would be useful.

- Fpockets results added to supplemental information as a new Supplemental Figure 1 with legend:

- **Supplementary Figure 1.** Fpocket result for the 3FKE crystal structure (gray) and for an open state from our MSM (blue) highlighting the opening of a druggable pocket. The groups of multicolored spheres are pockets computed using fpocket <https://github.com/Discngine/fpocket> which captures pockets in protein structures. In A) we applied the fpocket algorithm to detect pockets in the VP35 crystal structure. The shown pockets all have druggability scores less than 0.5 excepting the pocket highlighted with white spheres which has a druggability score of 0.578 and is known binding site for an inhibitor of nucleoprotein-VP35 interaction. B) Fpocket applied to a highly populated, open state from our MSM. The green spheres highlight a pocket where our methods also report a pocket. The druggability score of this pocket is 0.681 higher than the proposed druggability cutoff of 0.5.
- We agree that introducing substitutions to pocket residues would be of use to test small molecule binding to that pocket. However, we have not yet identified ligands that non-covalently bind to this pocket. The aim of this work is to elucidate the structural ensemble available to VP35. This is a common approach in structural studies, to first identify structural states then study ligand binding separately. As such, we think our work remains complete for this publication while agreeing that future studies including substitutions to residues in the pocket with potential ligands are important and helpful.

(2) Identification of a cryptic allosteric site: How can this be defined a binding "site"? The statement that the authors discovered a potential allosteric site appearing weak, without an actual proof to target this site with a small molecule. At least a docking analysis could be provided in the supporting information.

Response: We thank the reviewer for pointing out this concern. We agree with the suggestion and have therefore updated main text figure 2 with a figure (panel E shown below) showing docked organic molecules to a pocket open state.

○ Updated figure 2 caption (underlined text is the new addition): Figure 2. Exposons identify a large cryptic pocket and suggest potential allosteric coupling. A) Structure of VP35's IID highlighting residues in two exposons (blue and orange), the N-terminus (N-term), and C-terminus (I340). B) Network representation of the coupling between the solvent exposure of residues in the two exposons. The edge width between residues is proportional to the mutual information between them. C) Structure highlighting the opening of a cryptic pocket via the displacement of helix 5 that gives rise to the blue exposon. D) Structure highlighting the conformational change that gives rise to the orange exposon overlaid on the crystal structure (gray) to highlight that the rearrangements are subtler than in the blue exposon. E) FTMap results for the main cryptic pocket as shown in C and hotspots where a variety of small organic probes (multicolored sticks) form energetically favorable interactions. The probe molecules are intended to capture different drug-like interactions (such as hydrogen bonding and Van der Waals contacts) and include acetamide, acetonitrile, acetone, acetaldehyde, methylamine, benzaldehyde, benzene, isobutanol, cyclohexane, N,N-dimethylformamide, dimethyl ether, ethanol, ethane, phenol, isopropanol, or urea.⁴⁹⁻⁵²

- We added a mention of FTMap in the results section, paragraph four, line 38 (addition in underlined text): Retrospective analysis of other validated drug targets suggests

cryptic sites created by the movement of secondary structure elements, such as the displacement of helix 5, are often druggable.⁴⁶ The potential druggability of this cryptic site is also supported by application of the Fpocket and FTMap algorithms.^{47,48} Fpocket predicts this cryptic site to have a high druggability score (0.681) and FTMap highlights a number of hotspots within the pocket where small molecules could form a variety of energetically-favorable interactions (Supplementary Fig. 1 and Fig. 2E). Unfortunately, disrupting backbone binding is of less therapeutic utility than disrupting blunt end binding and it is unknown whether the contacts between A306, K309, and S310 are essential for backbone binding. Therefore, it is unclear from this analysis alone whether drugging this newly discovered cryptic pocket would be useful.

(3) My second concern is about the simulated systems. Authors used the apo VP35 protein (PDB ID: 3FKE) in their MD simulations. This PDB structure of VP35 is a homodimer without dsRNA. However, in the crystal structure of VP35-dsRNA complex (PDB ID: 3L25), the structure of VP35 is a homotetramer. The authors should claim the VP35 homodimer physiologically existed in virus. If the VP35 protein is in the homotetramer form in the physiological state, the construction of the simulated systems in the current study was artificial. Furthermore, I suggest authors using PDB 3L25 to perform additional simulations, in which dsRNA was removed from the structural complex to simulate the homotetramer. Authors should validate whether the predicted cryptic allosteric site in the homodimer protein reproduced in the homotetramer protein.

Response: We thank the reviewer for their observation about the macromolecular state of the protein in the crystal structures containing RNA. Additionally, previous studies (Leung, D. W., et al. (2009). PNAS **106**(2): 411-416. Fig S2) showed that biochemically purified VP35 is monomeric in solution supporting our choice of simulation set up for proper comparison to experimental results.

- We seeded simulations of 3L25 as suggested. We find that the homotetramer dissociates during equilibration, validating our ‘monomeric’ approach.
- We added a statement clarifying our choice in system setup in the methods section lines 5-6 (underlined text):
 - Simulations were initiated from the apo protein model of PDB 3FKE^{33,34} and run with Gromacs⁶⁰ using the amber03 force field⁶¹ and TIP3P explicit solvent⁶² at a temperature of 300 K and 1 bar pressure, as described previously.⁶³ Recombinant VP35 IID is known to be monomeric supporting our choice in system setup. We first applied our FAST-pockets algorithm⁴³ to balance 1) preferentially simulating structures with large pocket volumes that may harbor cryptic pockets with 2) broad exploration of conformational space. For FAST, we performed 10 rounds of simulations with 10 simulations/round and 80 ns/simulation. To acquire better statistics across the landscape, we performed an RMSD-based clustering using a hybrid k-centers/k-medoids algorithm⁶⁴ implemented in Enspara⁶⁵ to divide the data into 1,000 clusters.

(4) The abstract and discussion sections regarding to the method development in the identification of cryptic allosteric sites are insufficient. Recent work from other groups should

be added.

Response: We appreciate this note and have updated our introduction to also include additional references that capture recent work from other groups.

- **Relevant Section:** Cryptic pockets present opportunities for designing drugs for difficult targets like PPIs and PNIs but identifying and exploiting these pockets remains challenging.⁴⁻⁶ Cryptic pockets are absent in available experimental structures but form in a subset of excited states that arise due to protein dynamics. These cryptic sites can serve as valuable drug targets if they coincide with key functional sites, or if they are allosterically coupled to distant functional sites.^{7,8} Most known cryptic sites were only identified after the serendipitous discovery of a small molecule that binds and stabilizes the open form of the pocket.^{8,9} However, one would ideally like to decouple pocket discovery from ligand discovery. Towards this end, we have developed a suite of computational and experimental methods for detecting cryptic pockets and allostery, in addition to other available approaches.^{10-12,13-24} We have successfully applied subsets of this toolset to a number of enzymes that are established drug targets,^{12,25} suggesting that the same tools may be ready for application to challenging targets like PPIs and PNIs.
- To the above underlined text, we added the following citations and their corresponding reference numbers.
 - [21] Ni et al., Chem Sci., 2021, 12, 464
 - [22] Lu et al., Ncomms, 2021, 12, 4721
 - [23] Raich et al., PNAS, 2021, 118 (4)
 - [24] Vajda et al., Curr. Opin. Chem. Biol., 2018, 44:1-8

(5) Movie S1 and Movie S2: The labels of key domains would facilitate the reader to interpret the movies. One could also think about some arrows that would further suggest the region of protein on which the reader should focus, according to Authors interpretation. In addition, in Figure 1, helix number should be labeled that would facilitate the reader to understand authors' structural description in the text.

- We thank the reviewer for noting where added clarity is needed. We labeled the helices in Figure 1 and updated the movies per these suggestions.
- For Supplemental Movie 1, we added labels for the N-terminus, the alpha helical and beta sheet subdomains, and a labeled arrow showing the direction of pocket opening and updated the movie caption (underlined text).

- Supplementary movie 1: A morph between the x-ray structure and an example open state of the blue exposon showing the large motion of helix 5 that opens the VP35 IID cryptic pocket. The movement of the helix on the beta sheet subdomain away from the alpha helix subdomain comprises pocket opening which the arrow highlights.
- For Supplemental Movie 2, we labeled the N-terminus, the alpha helical and beta sheet subdomains, and used an arrow to highlight the concerted motion that does not open up a pocket and updated the movie caption (underlined text).

- Supplementary movie 2: A morph between the x-ray structure and an example open state of the orange exposon showing the motion that gives rise to this exposon. The labels note the location of the N-terminus, and the two subdomains, the shown arrow highlights the movement of loops that comprise the orange exposon.

Reviewer #2 (Remarks to the Author):

Disrupting protein-nucleic acid interactions (PNIs) with small molecules is a promising therapeutic strategy. Unfortunately, most known PNIs lack evident druggable pockets and are thus considered intractable targets.

Cryptic (hidden) pockets might provide a valid alternative approach for designing drugs targeting PNIs. However, by their nature cryptic pockets are not visible in the unliganded (apo) crystal structure of the target proteins. Methods based on enhanced sampling molecular dynamics simulations can help in this respect. In this paper the authors show how a combination of adaptive sampling molecular dynamics simulations with thiol labeling and mutagenesis experiments can be used to successfully identify a cryptic allosteric pocket allosterically coupled to a protein-nucleic acid interface.

The chosen protein, the interferon inhibitory domain (IID) of Ebola viral protein 35 (VP35), plays a crucial role in antagonizing the host's innate immunity and thus is a promising drug target. However, crystal structures have shown that the binding interfaces of VP35 IID with dsRNA are flat and lack evident cavities. This makes it difficult to design small molecules that bind at the interface and disrupt the PNIs with classic computer aided approaches.

In contrast to classic (static approaches) adaptive sampling simulations and machine learning algorithms are able predict a cryptic pocket that is allosterically coupled to a key dsRNA-binding interface. By performing thiol labeling and mutagenesis experiments the authors are able to convincingly validate the predicted pocket, while covalent modifications that mimic drug binding at the cryptic pocket successfully disrupt dsRNA binding.

The successful identification of a cryptic pocket allosterically coupled to a PNIs shows the power of predictive protein dynamics simulations in the rational targeting of 'intractable' proteins and paves the avenue for the use of combined computational and experimental pipelines to target other difficult PNIs.

- We thank the reviewer for their support.

Reviewers' Comments:

Reviewer #1:

Remarks to the Author:

A more comment should be emphasized:

The authors have identified the cryptic allosteric pocket of VP35 in silico adequately and tried to verify the site with CD and fluorescence experiments. Unfortunately, except docking analysis with some nonspecific binding organic probes, there is no active allosteric compound and no more experiment such as complex crystal or cryo-EM structure to confirm the allosterically control function.